ⓐ | Open Peer Review | Human Microbiome | Research Article

# Tezacaftor/Ivacaftor therapy has negligible effects on the cystic fibrosis gut microbiome

Ryan Marsh,[1] Claudio Dos Santos,[2] Liam Hanson,[2,3] Christabella Ng,[4,5] Giles Major,[4,6] Alan R. Smyth,[4,5] Damian Rivett,[2] Christopher van der Gast[1,7]

**ABSTRACT** People with cystic fibrosis (pwCF) experience a range of persistent gastrointestinal symptoms throughout life. There is evidence indicating interaction between the microbiota and gut pathophysiology in CF. However, there is a paucity of knowledge on the potential effects of CF transmembrane conductance regulator (CFTR) modulator therapies on the gut microbiome. In a pilot study, we investigated the impact of Tezacaftor/Ivacaftor dual combination CFTR modulator therapy on the gut microbiota and metabolomic functioning in pwCF. Fecal samples from 12 pwCF taken at baseline and following placebo or Tezacaftor/Ivacaftor administration were subjected to microbiota sequencing and to targeted metabolomics to assess the short-chain fatty acid (SCFA) composition. Ten healthy matched controls were included as a comparison. Inflammatory calprotectin levels and patient symptoms were also investigated. No significant differences were observed in overall gut microbiota characteristics between any of the study stages, extended also across intestinal inflammation, gut symptoms, and SCFA-targeted metabolomics. However, microbiota and SCFA metabolomic compositions, in pwCF, were significantly different from controls in all study treatment stages. CFTR modulator therapy with Tezacaftor/Ivacaftor had negligible effects on both the gut microbiota and SCFA composition across the course of the study and did not alter toward compositions observed in healthy controls. Future longitudinal CFTR modulator studies will investigate more effective CFTR modulators and should use prolonged sampling periods, to determine whether longer-term changes occur in the CF gut microbiome.

**IMPORTANCE** People with cystic fibrosis (pwCF) experience persistent gastrointestinal (GI) symptoms throughout life. The research question "how can we relieve gastrointestinal symptoms, such as stomach pain, bloating, and nausea?" remains a top priority for clinical research in CF. While CF transmembrane conductance regulator (CFTR) modulator therapies are understood to correct underlying issues of CF disease and increasing the numbers of pwCF are now receiving some form of CFTR modulator treatment. It is not known how these therapies affect the gut microbiome or GI system. In this pilot study, we investigated, for the first time, effects of the dual combination CFTR modulator medicine, Tezacaftor/Ivacaftor. We found it had negligible effects on patient GI symptoms, intestinal inflammation, or gut microbiome composition and functioning. Our findings are important as they fill important knowledge gaps on the relative effectiveness of these widely used treatments. We are now investigating triple combination CFTR modulators with prolonged sampling periods.

**KEYWORDS** gut microbiome, gut microbiota, dysbiosis, CFTR modulator therapy

People with cystic fibrosis (pwCF) experience a range of persistent gastrointestinal (GI) abnormalities that affect quality of life. The research question "how can we relieve gastrointestinal symptoms, such as stomach pain, bloating, and nausea?" continues to be

Address correspondence to Christopher van der Gast, chris.vandergast@northumbria.ac.uk, or Damian Rivett, d.rivett@mmu.ac.uk.

R.J.M., C.D.S., and L.H. have nothing to disclose. D.R. and C.v.d.G. report grant funding from Vertex Pharmaceuticals outside of the submitted work. C.N., G.M., and A.R.S. report grants and speaker honorarium from Vertex Pharmaceuticals outside the submitted work. G.M. is an employee of Société Produits de Nestlé S.A.

See the funding table on p. 11.

among the top priorities for clinical research in CF (1, 2). Also present with GI abnormalities is dysbiosis of the gut microbiome, which are changes to the resident microbiota and their functional outputs that are hypothesized to exacerbate abnormalities associated with CFTR dysfunction (3). Indeed, changes to the gut microbiota have previously been associated with markers of intestinal function, inflammation, and local tissue damage (4, 5), indicating a role of the microbiome in the multifactorial etiology of intestinal disease in the CF gut. We have previously demonstrated such relationships between the gut microbiota composition and intestinal function in pwCF, including markers of gut pathophysiology and intestinal function, as measured by magnetic resonance imaging (6).

Over two-thirds of pwCF in the UK are now receiving CF transmembrane conductance regulator (CFTR) modulator therapies (7), which can correct the underlying problems of mutated CFTR protein functioning, including trafficking, gating, and conductance at the cell epithelial surface (8). In comparison to the lower airways, our knowledge of the effects of CFTR modulator therapies on CF GI pathophysiology is limited. While improvements to BMI, patient growth rates, and intestinal pH increases are better defined (9–12), there remain differing results concerning the modulation of intestinal inflammation from the small number of available studies, all of which were focused on Ivacaftor or Lumacaftor/Ivacaftor CFTR modulator-based treatment (13–15). Likewise, studies investigating changes to the microbiota are also scarce and relate mostly to monotherapy approaches (13, 14, 16, 17). The CF gut microbiota is currently divergent from healthy controls throughout life and further compounded by CF-associated lifestyle factors such as antibiotic therapies and dietary changes (18–22). It is, however, reasonable to suggest restoration of CFTR function could remodulate the bacterial composition back to a signature observed in healthy controls, given that the primary consequence of CFTR dysfunction alone is sufficient to induce dysbiosis in the CF population (23). This may arise from the restoration of fluidity at the site of the intestinal epithelia, or through various other pathways and mechanisms in which the CFTR protein plays a key role (24).

A new clinical research priority for CF is to understand "what are the effects of modulators on systems outside the lungs such as … gastrointestinal?" (25). As new CFTR modulator therapies become available it is important to understand potential impacts on the GI system, including the gut microbiome (5). Therefore, in the current pilot study, we aimed to investigate the impact of Tezacaftor/Ivacaftor (Tez/Iva) dual combination CFTR modulator therapy on the gut microbiome in pwCF. This was achieved by analyzing and comparing the gut microbiota along with short-chain fatty acid (SCFA) targeted metabolomics from fecal samples taken from pwCF at baseline and following placebo or Tezacaftor/Ivacaftor dual combination CFTR modulator therapy. These samples were collected as part of a randomized crossover trial of Tezacaftor/Ivacaftor versus placebo (NCT04006873). Fecal samples from healthy matched controls were included as a comparison (6). SCFAs were specifically targeted as these microbially produced metabolites are known to play important roles in gut health and development of disease (26). Patient clinical characteristics at baseline are detailed in Table 1.

## RESULTS

Bacterial taxa within the whole microbiota were partitioned into common core and rare satellite taxa after plotting distribution-abundance relationships for all sample groups from baseline, placebo, and Tezacaftor/Ivacaftor treatment periods, and the healthy controls (Figure S1). Core taxa within each treatment period along with the healthy control group are given in Table S1. Diversity and composition for the whole microbiota, as well as the core and satellite taxa, between treatment periods were compared (Fig. 1A and B).

No significant differences were observed in whole microbiota diversity between any of the treatment periods ($P > 0.05$ in all instances) (Fig. 1A; Table S2). Significant differences in diversity were observed in core taxa between baseline and placebo ($P = 0.007$) and placebo and Tezacaftor/Ivacaftor treatment groups ($P = 0.039$). Core taxa diversity was

**TABLE 1** Clinical characteristics during Tezacaftor/Ivacaftor trial period in pwCF[a]

| Characteristic | |
|---|---|
| Baseline age (mean ± SD) | 20.8 ± 7.8 |
| Male (%) | 8 (66.6) |
| Baseline BMI (mean ± SD) | 21.4 ± 2.5 |
| p.Phe508del/p.Phe508del (%) | 12 (100) |
| Pancreatic insufficient (%) | 12 (100) |
| Baseline FEV1% (mean ± SD) | 80.6 ± 19.6 |
| Regular antibiotics (%) | 11 (91.6) |
| Additional antibiotics during trial (%) | 6 (50) |

[a]Regular antibiotics during trial include: oral azithromycin (7/12, 58.3%); inhaled tobramycin (2/12, 16.7%); inhaled aztreonam (1/12, 8.3%); and inhaled colistimethate sodium (4/12, 33.3%). Additional antibiotic treatment includes: oral ciprofloxacin (4/12, 33.3%); oral clarithromycin (1/12, 8.3%): and intravenous (IV) antibiotics (1/12, 8.3%).

not significantly different between baseline and Tezacaftor/Ivacaftor groups ($P > 0.05$) (Fig. 1A; Table S2). For the satellite taxa, no significant differences in diversity were observed between treatment periods ($P > 0.05$ in all instances) (Fig. 1A; Table S2). Additionally, composition of the whole microbiota and the core and satellite taxa groups was not significantly different between treatment periods ($P > 0.05$ in all instances) (Fig. 1B; Table S3).

Next diversity and composition of the microbiota, core taxa, and the satellite taxa between each treatment period and the matched healthy control group were compared (Fig. 1C and D). In all instances, diversity in the control group was found to be significantly higher when compared to each treatment period ($P < 0.05$ in all instances) (Fig. 1C; Table S4). Similarly, the microbiota, core taxa, and satellite taxa compositions of the healthy control samples were significantly different from pwCF for all treatment periods ($P < 0.05$ in all instances) (Fig. 1D; Table S5). To visualize how Tezacaftor/Ivacaftor treatment might shift the microbiota composition back to that observed in healthy controls, samples were spatially plotted utilizing Bray-Curtis distances (Fig. 2). Healthy control samples clustered more tightly to one another, indicating they were more similar to each other, when compared to samples within any of the treatment periods (Fig. 1D and 2). Also, the healthy control microbiota samples clustered away from the treatment stage microbiota samples, which all overlapped with one another. No shift within the Tezacaftor/Ivacaftor group back toward a healthy microbiota composition was observed (Fig. 2).

Changes in microbially produced SCFA metabolites were also investigated (Fig. 3). SCFA metabolites, including their target and confirmation ions, are listed in Table S6. SCFA metabolite compositions were not significantly different between treatment stage sample groups ($P > 0.05$ in all instances) (Table S7). Conversely, significant differences in SCFA compositions between healthy controls and all treatment stages were observed ($P < 0.05$ in all instances) (Table S7). In terms of specific SCFAs, similarity of percentages (SIMPER) analysis (Table S8) showed that acetic, propionic, and butyric acid cumulatively contributed >73% of these differences. Furthermore, their combined relative levels were increased in pwCF compared to healthy controls, which constituted mean (±SD) collective levels of 92.5% (±6.6%) and 84.3% (±6.6%), respectively. This coincided with higher relative levels of longer SCFAs in healthy controls, particularly those containing ≥5 carbons, of which there were significant differences compared to pwCF, as seen in Table S9. Additionally, isobutryic acid relative levels were significantly decreased in both baseline ($P = 0.048$) and placebo ($P = 0.009$), but not Tezacaftor/Ivacaftor ($P = 0.429$) pwCF samples compared with healthy controls (Table S9).

Finally, no significant differences in intestinal inflammation, as measured by fecal calprotectin, were observed between Tezacaftor/Ivacaftor and baseline/placebo phases of the study [median (IQR): 13.7 (5.2–25.8) vs 12.6 (8.0–21.5) µg/g, $P = 0.954$]. However, both phases were significantly different from the healthy controls [3.7 (2.8–4.8) µg/g, $P = 0.010$ and $P = 0.018$, respectively]. Also, no differences across participant symptom scores

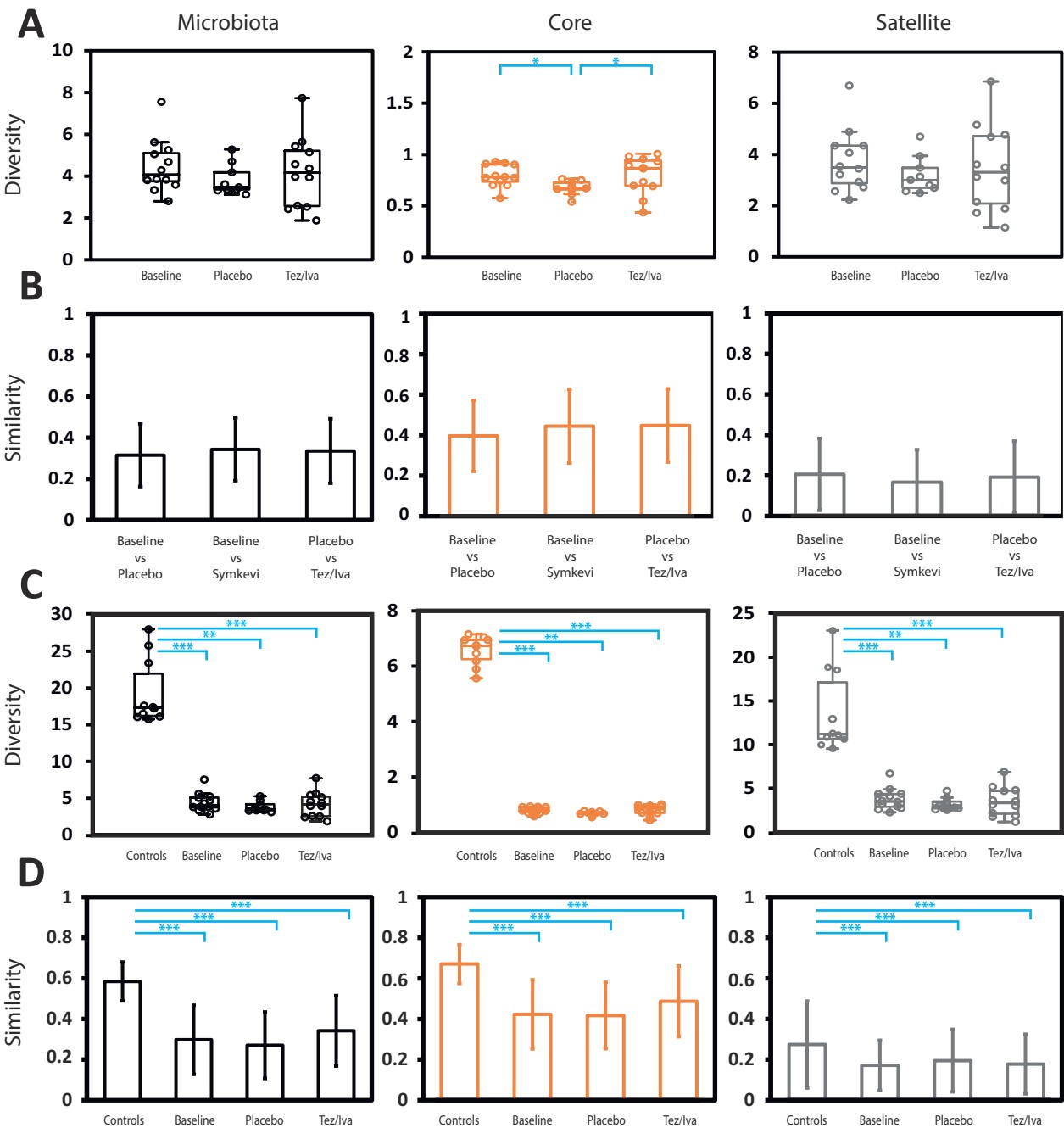

**FIG 1** Comparisons of microbiota diversity and similarity indices. (A) Differences in Fisher's alpha index of diversity across pwCF during the various treatment periods. Circles indicate individual patient data for microbiota (black), partitioned core (orange), and partitioned satellite (gray) taxa. Error bars represent 1.5 times the interquartile range (IQR). Asterisks denote significant differences in diversity between treatment periods following Kruskal-Wallis testing. Summary statistics are provided in Table S2. (B) Microbiota variation measured across various treatment periods, utilizing the Bray-Curtis index. Shown is the similarity between different treatment periods. Error bars represent standard deviation of the mean. Analysis of similarities (ANOSIM) tests were conducted between sampling phases. Summary statistics are provided in Table S3. (C) Differences in Fisher's alpha index of diversity across pwCF from this trial compared to previously matched healthy controls. Asterisks denote significant differences in diversity between treatment periods following Kruskal-Wallis testing. Summary statistics are provided in Table S4. (D) Microbiome variation across the various treatment periods in pwCF and matched healthy controls, utilizing the Bray-Curtis index. Shown is the within-group similarity between different treatment periods. Error bars represent standard deviation of the mean. Asterisks denote significant differences following ANOSIM testing. Summary statistics are provided in Table S5. ***; $P < 0.0001$, **; $P < 0.001$, *; $P < 0.05$. Group sizes: Baseline, $n = 12$; placebo, $n = 9$; and Tez/Iva, $n = 12$.

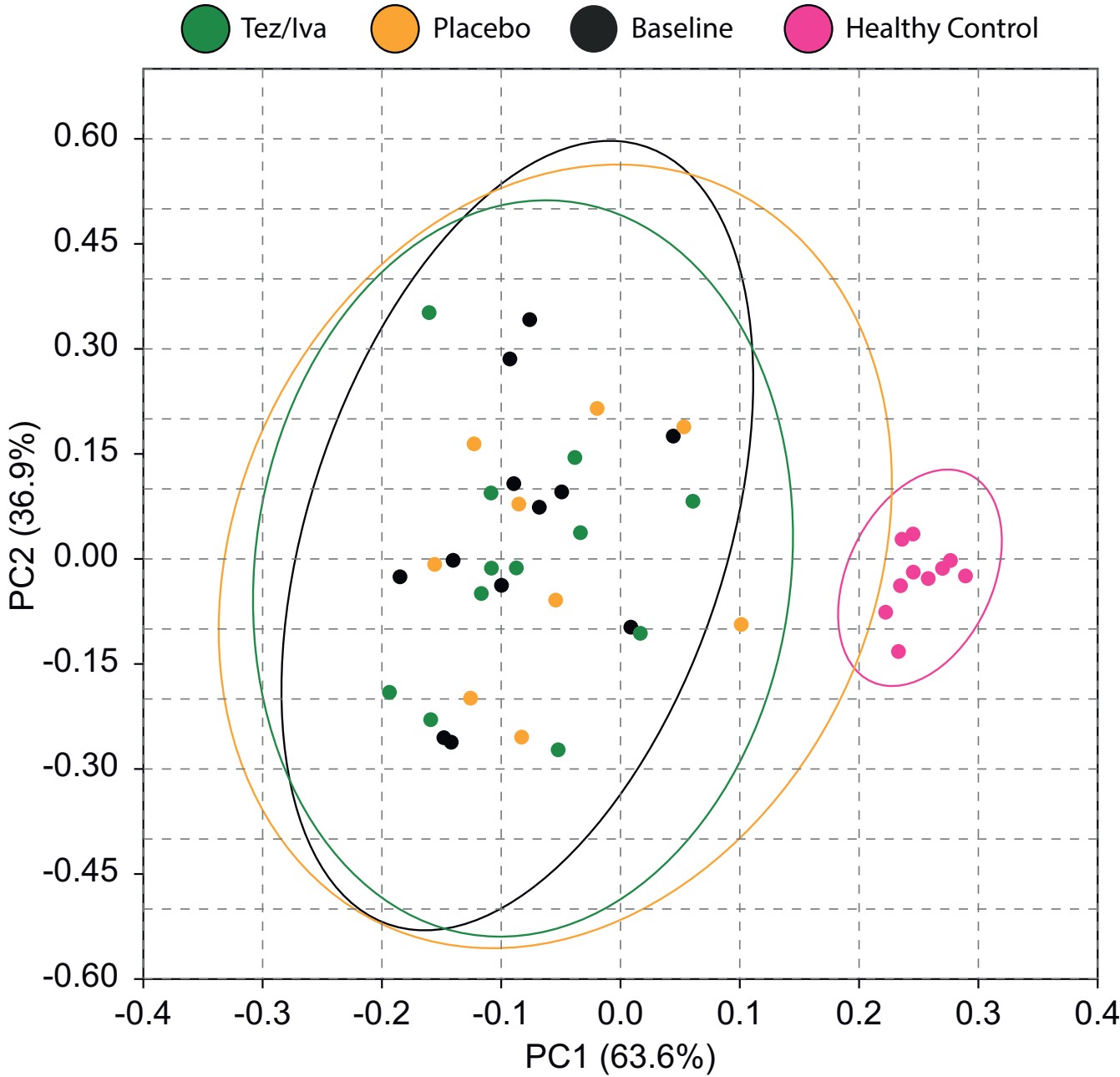

**FIG 2** Principle coordinates analysis of gut microbiota composition from different treatment periods, and also matched healthy controls, utilizing Bray-Curtis distances. Each data point represents an individual sample. Ellipses represent a 95% confidence level between groups. Color of data point is indicative of group as depicted. ANOSIM statistics for bacterial compositions between treatment periods and comparisons with healthy controls are found in Tables S3 and S5, respectively. Group sizes: Baseline, $n = 12$; placebo, $n = 9$; and Tez/Iva, $n = 12$. ANOSIM, Analysis of similarities.

between Tezacaftor/Ivacaftor and off-treatment samples through the PAC-SYM ($P = 0.393$) or CFAbd ($P = 0.297$) questionnaires were observed (Table S10 and S11).

## DISCUSSION

As new CFTR modulator therapies become available to greater numbers of eligible pwCF, it is crucial to investigate potential treatment effects on not just the lungs but a wide range of systems in CF, including the GI system (6, 25). To the best of our knowledge, this is the first study investigating the impact of Tezacaftor/Ivacaftor on the gut microbiome in pwCF. Here, we examined the impact of Tezacaftor/Ivacaftor administration on the gut

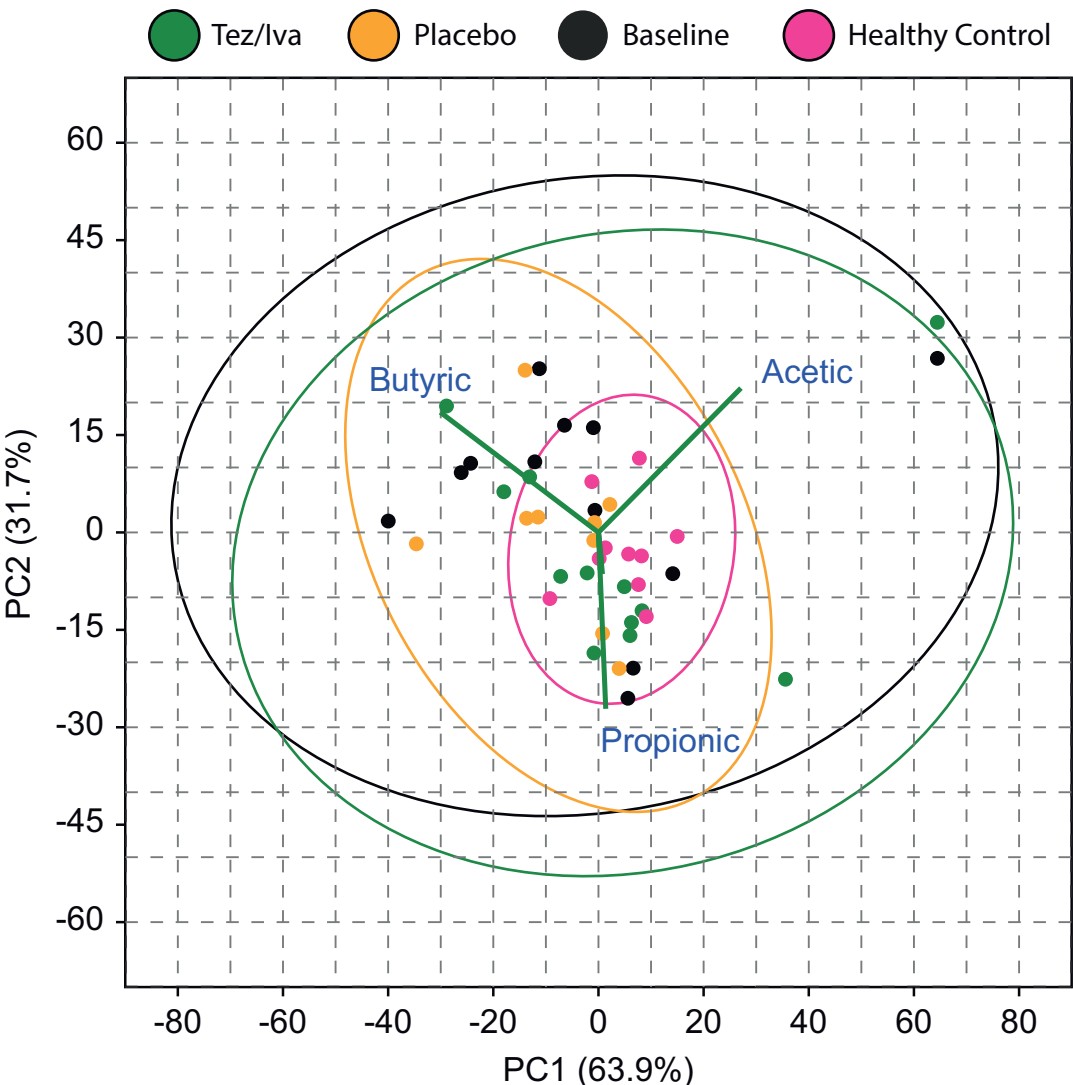

**FIG 3** Principle component analysis (PCA) plot of the SCFA (C2–C7) profiles of grouped samples at baseline, placebo, and treatment periods. Also included are healthy control subjects for comparison. Ellipses represent a 95% confidence level between groups. Color of data point is indicative of group as depicted. Summary statistics are found in the supplementary results, including analysis of similarities testing (Table S7), SIMPER analysis (Table S8), and Kruskal-Wallis testing of individual SCFAs (Table S9).

microbiota composition and metabolomic functioning by means of 16S rRNA targeted amplicon sequencing and targeted SCFA metabolomics. Our results indicate that Tezacaftor/Ivacaftor treatment had negligible effects on gut microbiota diversity and no significant differences on microbiota composition when compared to baseline or placebo treatment periods. There were differences across the core taxa observed within the placebo treatment period; however, the similar magnitude of Fisher's alpha likely indicates these changes render little biological significance and are resultant of alternate temporal variation. Furthermore, pwCF samples across all treatment periods exhibited significantly reduced diversity and intra-group similarity, alongside a significantly different composition of bacterial taxa compared to matched healthy controls, suggestive of a perturbated community that is commonly observed in the CF gut (6, 19, 20, 22).

Previous studies investigating the effects of CFTR modulator therapy on the gut microbiota have mainly been limited to Ivacaftor treatment, where the majority of studies observed no differences across bacterial diversity or overall composition following Ivacaftor usage (13, 14, 16). While Kristensen et al. did observe significant

changes to the aforementioned (17), this was only evident following extended 12 months of treatment in pwCF harboring the p.Ser1251Asn mutation. Given the increased efficacy of Ivacaftor for pwCF undergoing treatment for class III mutations relative to those administered dual therapy to treat p.Phe508del in the respiratory domain (27), it may be possible that this reduced efficacy also extends to the site of the intestinal tract and changes to the microbiota are more subtle in such cohorts of pwCF. This will be clarified as further studies on CFTR modulators and the intestinal microbiome encompass greater numbers of participants.

Additionally, Pope et al. investigated dual-combination Lumacaftor/Ivacaftor treatment in a p.Phe508del cohort (16) but did not observe any effects on bacterial diversity or composition following. Heterogeneity across studies published thus far includes clinical and patient characteristics that have previously been shown to impact or associate with microbiota composition, including CFTR genotypes (23, 28, 29), patient age (19, 30), sex (6), and varying antibiotic regimens (4, 20, 22). These factors, alongside variable pancreatic function across patients, should be considered in the wider context of the different outcomes observed.

Our cohort had an overall (mean ± SD) lung function (FEV$_1$ %) of 80.6 ± 19.6 at baseline, the largest value we are aware of for a CFTR modulator-based gut microbiota study. Future studies may elucidate the relevance of baseline respiratory characteristics toward gut microbiota changes following CFTR modulator usage, given the strong evidence for a gut-lung axis in CF (20, 31). Alongside others, we have previously demonstrated the large association of antibiotic therapy with microbiota composition across pwCF (6, 18, 20), and in the current study, all but one patient was receiving regular antibiotic therapy as part of their routine care. Additionally, 50% (6/12) of pwCF were administered additional antibiotic therapy either during or recently preceding their fecal samples and clinical assessments.

With regard to the functionality of the microbiota, we did not observe any differences in the fecal relative levels of SCFAs between any of the treatment periods in this study. Although only approximately 5% of SCFAs are excreted in feces (32, 33), fecal levels of SCFAs have been shown to relate to disease severity and patient symptoms across inflammatory bowel diseases (IBDs) (34–36) and therefore may be of use in the CF gut as biomarkers of microbiota functional capacity. This is unsurprising, given the relationship of butyrate in particular with anti-inflammatory properties and enhancement of epithelial integrity (37–39). Fecal levels of prominent SCFAs, including acetic, propionic, and butyric acids, have previously been shown to be lower in the CF gut compared to healthy controls (40). When extending our analyses to matched healthy controls, we found significant differences in composition compared to pwCF. While compositional differences were mainly driven by levels of acetic, propionic, and butyric acids, subsequent univariate analyses between pwCF and healthy controls revealed differences in the relative levels of longer-chain SCFAs. Isobutyric relative levels were significantly increased in controls compared to baseline and placebo, but not Tezacaftor/Ivacaftor pwCF samples, which could reflect temporal changes such as increased protein availability and subsequent amino acid fermentation resulting from dietary fluctuations (41). We detected that valeric, hexanoic, and heptanoic acids were significantly increased in healthy control subjects compared to pwCF. While these longer-chain fatty acids have also been implicated in IBD (42), future investigation should adopt extensive integrative approaches to better understand relationships between the microbiome, metabolome, and intestinal clinical outcomes further. This may elucidate any potential functional redundancy of the microbiota (43) more clearly in the context of CFTR modulator treatment that is likely administered in the presence of antibiotic therapies and other lifestyle alterations that persist within CF.

Persistent intestinal abnormalities and symptoms are a hallmark of CF gastrointestinal disease, including intestinal inflammation, for which we identified significant differences between pwCF and healthy controls, but not across the various treatment periods in pwCF. The latter is similar to Ronan et al. (13), but contrary to results from others (9,

14, 15), further suggesting intestinal inflammation in CF is multi-factorial by nature. The lower fecal calprotectin values obtained in our study (compared to other CF studies) during all treatment phases suggest that the participants in our study had less gut inflammation. A cutoff value of <50 µg/g is generally used to define normal levels (44). A more comprehensive approach may therefore be required to determine the severity of intestinal inflammation in future studies. Small intestinal bacterial overgrowth is another common abnormality of the CF gut that has so far persisted during interventions with Lumacaftor/Ivacaftor therapy (45). As it is often related to increased oro-caecal transit times (46), it will be interesting to determine if Tezacaftor/Ivacaftor impacts such gut function metrics, given the relationships with the microbiota we have previously described in pwCF (6).

While we observed no differences across intestinal symptoms in our cohort through the CFAbd and PAC-SYM questionnaires, it is anticipated that more recent triple combination therapies, such as Elexacaftor-Lumicaftor-Ivacaftor (ETI), may alleviate the GI manifestations and symptoms of CF. Indeed, preliminary data surrounding its usage and patient symptoms are promising, based on the reduction of symptoms across the CFAbd-scores reported by Mainz et al. (47). Triple combination therapy (for patients with at least one copy of the common p.Phe508del mutation) leads to fewer pulmonary exacerbations and overall improved respiratory health (48). This may in turn allow for a reduction in antibiotic use and a re-shaping of the gut microbiota, so that it resembles more closely signatures observed in healthy controls. It is also logical to postulate that the initiation of CFTR modulator treatment earlier in life will also increase microbiota similarity between pwCF and the wider population, particularly if other GI manifestations are minimized. Should this not arise, despite the patient clinical improvement, the further integration of multi-omic approaches will likely clarify if the predisposed microbial community exhibit changes to functionality that promote a favorable intestinal environment for pwCF. Finally, ETI therapy demonstrates increased efficacy across intestinal epithelia as compared to Tez/Iva in biopsies from pwCF homozygous for the p.Phe508del mutation, which our current cohort all exhibited (49). As we obtain deeper knowledge surrounding triple-therapy modulator usage, the findings of this study should, therefore, contribute valuable insights into the complex challenge of comprehending the associations between restoring CFTR functionality and alterations to the intestinal microbiota.

A limitation of this pilot study is inevitably the small sample size of our cohort, which limits the power of specific analyses and restricts the ability to investigate confidently the effects of the various antibiotic regimens (antibiotic class, dosage, and frequency) across our patients. While the treatment period was also relatively short, longer-term administration has previously failed to elicit changes to the intestinal microbiota across pwCF with similar genotypes (16); however, this does not include Tez/Iva administration. The double-blind crossover element of our study was limited to 9/12 (75%) of participants due to disruption from the COVID-19 pandemic and patient desires to switch to available open-label treatment, although samples at baseline and during Tezacaftor/Ivacaftor treatment were obtained from all participants. The principle strength of this study is the important first insights gained into the efficacy of Tezacaftor/Ivacaftor treatment in modulating the gut microbiota and its potential metabolomic capacity in a clinically representative cohort of pwCF harboring the p.Phe508del mutation. Our future work will look to encompass both respiratory and intestinal microbiota analyses, alongside extensive gut function metrics, and absolute quantification of microbiota-derived metabolites, following CFTR modulator therapy.

## Conclusions

This crossover pilot study has revealed no significant impact of Tezacaftor/Ivacaftor administration on gut microbiota composition or relative levels of fecal SCFAs within pwCF. Compositionally, the microbiota of pwCF is still very much distinct compared to that of healthy controls, demonstrating a lack of remodulation of the gut microbiome

by modulator therapy. The negligible effects observed in this study may be related to the short administration period of Tezacaftor/Ivacaftor, alongside other characteristics of pwCF, including continuous antibiotic treatment and sustained pancreatic insufficiency. Future studies with more efficacious CFTR modulators may elucidate the impact of modulating CFTR function, and implications of the CF lifestyle, on the microbiota more clearly.

## MATERIALS AND METHODS

### Study participants and design

Fourteen pwCF homozygous for p.Phe508del were initially recruited from Nottingham University Hospitals NHS Trust, with fecal samples ultimately available for analysis from 12 pwCF. These CF participants were enrolled in a randomized, double-blind, placebo-controlled crossover trial with Tezacaftor/Ivacaftor. Treatments were administered for 28 days with an intermediate 28-day washout period. At baseline, and between days 19 and 23 of each phase of treatment, participants attended the clinic to provide fecal samples and have clinical assessments undertaken, including completion of the validated PAC-SYM and CFAbd-Score questionnaires to assess gut symptoms (50, 51). Additionally, fecal samples from 10 age-matched healthy controls from our previous study were available for microbiota and metabolomic comparison (6). The full study design is described in the Supplementary Materials. Patient clinical characteristics at baseline are detailed in Table 1. Written informed consent, or parental consent and assent for pediatric participants, was obtained from all participants. Study approval was obtained from the UK National Research Ethics Committee (19/WM/0130). All fecal samples obtained were immediately stored at −80°C prior to processing for microbiota sequencing and metabolomics to reduce changes before downstream community analysis (52).

### Targeted amplicon sequencing

DNA from dead or damaged cells, as well as extracellular DNA was excluded from analysis via cross-linking with propidium monoazide prior to DNA extraction, as previously described (53). Cellular pellets resuspended in phosphate-buffered saline were loaded into the ZYMO Quick-DNA Fecal/Soil Microbe Miniprep Kit (Cambridge Bioscience, Cambridge, UK), as per the manufacturer's instructions. Dual mechanical-chemical sample disruption was performed using the FastPrep-24 5G instrument (MP Biomedicals, California, USA). Following DNA extraction, approximately 20 ng of template DNA was then amplified using Q5 high-fidelity DNA polymerase (New England Biolabs, Hitchin, UK) using a paired-end sequencing approach targeting the bacterial 16S rRNA gene region (V4–V5) as previously described (6). Pooled barcoded amplicon libraries were sequenced on the Illumina MiSeq platform (V3 Chemistry). Extended methodology, primers, and PCR conditions can be found in the Supplementary Materials.

### Sequence processing and analysis

Sequence processing and data analysis were initially carried out in R (version 4.0.1), utilizing the package DADA2 (54). The full protocol is detailed in the Supplementary Materials. Raw sequence data reported in this study have been deposited in the European Nucleotide Archive under the study accession number PRJEB57754.

### Gas-chromatography mass-spectrometry of fecal samples to investigate SCFA levels

Gas-chromatography mass-spectrometry (GC-MS) analysis was carried out using an Agilent 7890B/5977 Single Quadrupole Mass Selective Detector (Agilent Technologies) equipped with a non-polar HP-5ms Ultra Inert capillary column (30 m × 0.25 mm ×

0.25 µm) (Agilent Technologies). In brief, fecal samples stored at −80°C were ground in liquid nitrogen before lysis and homogenization in MS-grade water using ZR Bashing-Bead Lysis Tubes (Cambridge Bioscience, UK), on the FastPrep-24 5G instrument (MP Biomedicals, CA, USA). The SCFA layer was obtained by sample mixing at 4°C, and centrifugation at 13,000 × $g$ for 30 min. The supernatant containing fecal SCFAs was removed and protonated with 5 M HCl before the addition of anhydrous diethyl (DE) for the liquid-liquid extraction, again involving incubation, mixing, and centrifugation. This process was repeated twice, with the respective DE layers containing fecal SCFAs then equally pooled in a new Eppendorf tube pre-loaded with $Na_2SO_4$ to remove any water prior to transfer into the GC-MS vial, before addition of 2N, O-bis(trimethyl-silyl) trifluoroacetamide to derivatize the samples. The GC vial was capped tightly, and samples were vortexed and then incubated for 3 h at 37°C before loading onto the GC-MS and injection with an Agilent 7693 Autosampler. MS grade water processed in parallel was used as a blank sample to correct the background. Selected ion monitoring (SIM) mode was used for subsequent analyses; all confirmation and target ions lists are summarized in Table S6. Agilent MassHunter workstation version B.07.00 programs were used to perform post-run analyses. A $^{13}$C-short chain fatty acids stool mixture (Merck Life Science, Poole, UK) was used as the internal standard to normalise all spectra obtained prior to analyses. Extended information surrounding sample processing, SCFA extraction, derivatization, and GC-MS parameters can be found in the Supplementary Materials.

## Fecal calprotectin measurement

Stool was extracted for downstream assays using the ScheBo Master Quick-Prep (ScheBo Biotech, Giessen, Germany), according to the manufacturer's instructions. Fecal calprotectin was analyzed using the BÜHLMANN fCAL ELISA (Bühlmann Laboratories Aktiengesellschaft, Schonenbuch, Switzerland), according to the manufacturer's protocol.

## Statistical analysis

Regression analyses, including calculated coefficients of determination ($r^2$), degrees of freedom (df), $F$-statistic, and significance values ($P$), were utilized for microbial partitioning into common core and rarer satellite groups and were calculated using XLSTAT v2021.1.1 (Addinsoft, Paris, France). Fisher's alpha index of diversity and the Bray-Curtis index of similarity were calculated using PAST v3.21 (55). Tests for significant differences in microbiota diversity were performed using Kruskal-Wallis in XLSTAT. Student's $t$-tests used to determine differences in metadata were also performed in XLSTAT. Analysis of similarities with Bonferroni correction was used to test for significance in microbiota and SCFA composition and was performed in PAST. SIMPER analysis, to determine which constituents drove compositional differences between groups, and was performed in PAST. Statistical significance for all tests was deemed at the $P < 0.05$ level.

## ACKNOWLEDGMENTS

This work was funded by CF Trust grant (VIA 77) awarded to CvdG. The wider "Gut Imaging for Function & Transit in Cystic Fibrosis Study 2" (GIFT-CF2) was supported with funding from a Cystic Fibrosis Trust grant (VIA 061), a Cystic Fibrosis Foundation award (Clinical Pilot and Feasibility Award SMYTH18A0-I), and a Vertex Pharmaceuticals Investigator-Initiated Study award (IIS-2018-106697).

C.v.d.G., D.R., A.R.S., G.M., and R.M. conceived the microbiome study. R.M. and L.H. performed microbiota sample processing and analysis. R.M. and C.D.S. carried out the metabolomic analysis. R.M., D.R., and C.v.d.G. performed the data and statistical analysis. C.N., G.M., and A.S. were responsible for sample collection, clinical care records, and documentation. R.J.M., C.N., G.M., and C.v.d.G verified the underlying data. R.J.M., D.R., and C.v.d.G. were responsible for the creation of the original draft of the manuscript. R.J.M., C.N., G.M., D.R., A.R.S., and C.v.d.G. contributed to the development of the final

manuscript. C.v.d.G. is the guarantor of this work. All authors read and approved the final manuscript.

R.J.M., C.D.S., and L.H. have nothing to disclose. D.R. and C.v.d.G. report grant funding from Vertex Pharmaceuticals outside of the submitted work. C.N., G.M., and A.R.S. report grants and speaker honorarium from Vertex Pharmaceuticals outside the submitted work.

## AUTHOR AFFILIATIONS

[1]Department of Applied Sciences, Northumbria University, Newcastle, United Kingdom
[2]Department of Natural Sciences, Manchester Metropolitan University, Manchester, United Kingdom
[3]Department of Life Sciences, Manchester Metropolitan University, Manchester, United Kingdom
[4]School of Medicine, University of Nottingham, Nottingham, United Kingdom
[5]NIHR Nottingham Biomedical Research Centre, Nottingham, United Kingdom
[6]Nestlé Institute of Health Sciences, Société des Produits Nestlé, Lausanne, Switzerland
[7]Department of Respiratory Medicine, Salford Royal NHS Foundation Trust, Salford, United Kingdom

## AUTHOR ORCIDs

Christopher van der Gast  http://orcid.org/0000-0003-1101-4048

## FUNDING

| Funder | Grant(s) | Author(s) |
| --- | --- | --- |
| Cystic Fibrosis Trust (CF) | VIA 077 | Damian Rivett |
| | | Christopher van der Gast |
| Cystic Fibrosis Trust (CF) | VIA 061 | Claudio Dos Santos |
| | | Liam Hanson |
| | | Christabella Ng |
| | | Giles Major |
| | | Alan R. Smyth |
| | | Damian Rivett |
| | | Christopher van der Gast |
| Cystic Fibrosis Foundation (CFF) | SMYTH18A0-I | Claudio Dos Santos |
| | | Liam Hanson |
| | | Christabella Ng |
| | | Giles Major |
| | | Alan R. Smyth |
| | | Damian Rivett |
| | | Christopher van der Gast |
| Vertex Pharmaceuticals | IIS-2018-106697 | Claudio Dos Santos |
| | | Liam Hanson |
| | | Christabella Ng |
| | | Giles Major |
| | | Alan R. Smyth |
| | | Damian Rivett |
| | | Christopher van der Gast |

## AUTHOR CONTRIBUTIONS

Ryan Marsh, Data curation, Formal analysis, Investigation, Methodology, Writing – original draft, Writing – review and editing | Claudio Dos Santos, Formal analysis, Methodology, Validation, Writing – original draft | Liam Hanson, Formal analysis, Investigation, Methodology, Writing – original draft | Christabella Ng, Investigation, Resources, Writing – original draft | Giles Major, Conceptualization, Investigation, Supervision, Writing – original draft | Alan R. Smyth, Conceptualization, Funding acquisition, Resources, Supervision, Writing – original draft | Damian Rivett, Conceptualization, Formal analysis, Funding acquisition, Investigation, Methodology, Project administration, Supervision, Writing – original draft, Writing – review and editing | Christopher van der Gast, Conceptualization, Data curation, Formal analysis, Funding acquisition, Investigation, Methodology, Project administration, Resources, Supervision, Validation, Writing – original draft, Writing – review and editing

## DATA AVAILABILITY

Raw sequence data reported in this study has been deposited in the European Nucleotide Archive under the study accession number PRJEB57754.

## ADDITIONAL FILES

The following material is available online.

### Supplemental Material

**Supplemental material (Spectrum.01175-23-s0001.pdf).** Supplemental methods and materials.

### Open Peer Review

**PEER REVIEW HISTORY (review-history.pdf).** An accounting of the reviewer comments and feedback.

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
