## [Reviewer comments · Microbiology Spectrum]

Microbiology Spectrum

Tezacaftor/Ivacaftor therapy has negligible effects on the cystic fibrosis gut microbiome

Ryan Marsh, Claudio Dos Santos, Liam Hanson, Christabella Ng, Giles Major, Alan Smyth, Damian Rivett, and Christopher van der Gast

Corresponding Author(s): Christopher van der Gast, Northumbria University, and Damian Rivett, Manchester Metropolitan University

Review Timeline:

Submission Date:	March 17, 2023
Editorial Decision:	May 14, 2023
Revision Received:	June 22, 2023
Accepted:	June 28, 2023

Editor: Joanna Goldberg

Reviewer(s): Disclosure of reviewer identity is with reference to reviewer comments included in decision letter(s). The following individuals involved in review of your submission have agreed to reveal their identity: Thomas H Hampton (Reviewer #1)

Transaction Report:

DOI: <https://doi.org/10.1128/spectrum.01175-23>

May 14, 2023

Prof. Christopher van der Gast
Manchester Metropolitan University
Department of Life Sciences
John Dalton Building
Chester Street
Manchester M1 5GD
United Kingdom

Re: Spectrum01175-23 (Tezacaftor/Ivacaftor therapy has negligible effects on the cystic fibrosis gut microbiome)

Dear Prof. Christopher van der Gast:

Thank you for submitting your manuscript to Microbiology Spectrum. Your manuscript has been reviewed by two experts and by me. Each of the reviewers was pleased with the paper, but both had some minor suggestions of additional comments/discussion that could be added to place your findings into the context of the field. I agree with these suggestions and hope you can easily add or address these as you submit a revised version of the paper.

Link Not Available

Sincerely,

Joanna Goldberg

Journals Department
Reviewer comments:

Reviewer #1 (Comments for the Author):

This is a nicely written and well-conceived report that should interest many in the cystic fibrosis community. It's a shame, of course, that Teza/Iva doesn't really do much to improve the intestinal microbiome and CF patient gut health. Maybe Elexa/Teza/Iva will.

By way of comments and suggestions, I would recommend following up on this statement on line 101: "Significant differences in diversity were observed in core taxa between baseline and placebo ($P = 0.007$) and placebo and Tezacaftor/Ivacaftor treatment groups ($P = 103\ 0.039$). I would mention that although these differences reach statistical significance, all three CF groups had values of Fisher's alpha of about .75 and observed differences would be of questionable biological significance.

Another suggestion I have is to add a sentence or two about why we might have predicted that Texa/Iva would reshape bacterial communities. For example, do you suspect that restoring chloride channel function in gut epithelial cells would improve gut hydration and thereby promote the growth of certain bacteria?

Reviewer #2 (Comments for the Author):

Thank you for the opportunity to review the submitted manuscript (Spectrum01175-23) by Ryan Marsh and colleagues, titled "Tezacaftor/Ivacaftor therapy has negligible effects on the cystic fibrosis gut microbiome". The study examined the impact of Tezacaftor/Ivacaftor treatment on the gut microbiota composition and metabolomic functionality in twelve people with cystic fibrosis (pwCF). The researchers partitioned bacterial taxa into common core and rare satellite taxa and compared diversity and composition for the whole microbiota, as well as the core and satellite taxa, between treatment periods and healthy controls. The study found that Tezacaftor/Ivacaftor treatment did not significantly affect the gut microbiota composition or diversity, nor did it impact the production of short-chain fatty acid (SCFA) metabolites. The study also found that the relative levels of acetic, propionic, and butyric acid were increased in pwCF compared to healthy controls, whereas longer SCFAs containing {greater than or equal to}5 carbons were more abundant in healthy controls. Finally, the study found no significant differences in intestinal inflammation or participant symptom scores between Tezacaftor/Ivacaftor and off-treatment samples. Overall, this study provides insight into the effects of Tezacaftor/Ivacaftor on the gut microbiota in pwCF. Overall, the study is scientifically and methodologically accurate and conclusion are well supported by the reported results. However, I find that the current study has some limitations, some that are already highlighted in the manuscript, that might require a bit more detail on prior the manuscript being considered suitable for publication in Microbiology Spectrum.

Points:

1. What about pwCF that did not receive Tezacaftor/Ivacaftor or any other modulator therapy? Would the trajectory in the gut microbiota composition be similar and relatively stable in pwCF that never have received modulator therapy? Is possible that maturation of the gut microbiome in pwCF is affected by early adaptation of the microbiota to the "abnormal" environment of the lower GI and high antibiotic pressure in general?
2. As this pilot study is relatively small, which limits the generalisability of the results. Is it possible that larger sample sizes could have revealed more significant differences in microbiota composition and SCFA metabolites?
3. The study did not investigate the long-term effects of Tezacaftor/Ivacaftor on the gut microbiota, as the treatment period was only 4 weeks. Would the authors be able to further elaborate on if longer treatment durations could result in more significant changes in microbiota composition and SCFA or other metabolites in general?
4. Studies have shown limited changes in airway microbiota following modulator therapy, despite clinical improvement (both for single [<https://doi.org/10.1513/AnnalsATS.201907-493OC>; <https://doi.org/10.1016/j.jcf.2020.12.023>], double therapy [<https://doi.org/10.1128/spectrum.02251-22>; <https://doi.org/10.1183/23120541.00731-2020>] or even triple therapy [<https://doi.org/10.1016/j.jcf.2021.11.003>]), could the author further elaborate on if we would have expected significant changes in the GI microbiota? Would this be more associated with putative changes in functional capacities members of the GI microbiota?
5. With many individuals now being moved onto triple therapy, would the authors be able to comment on this in context to the current study and the importance of their findings?

Staff Comments:

Preparing Revision Guidelines

- Point-by-point responses to the issues raised by the reviewers in a file named "Response to Reviewers," NOT IN YOUR COVER LETTER.
- Upload a compare copy of the manuscript (without figures) as a "Marked-Up Manuscript" file.
- Each figure must be uploaded as a separate file, and any multipanel figures must be assembled into one file.

- Manuscript: A .DOC version of the revised manuscript
- Figures: Editable, high-resolution, individual figure files are required at revision, TIFF or EPS files are preferred

Please return the manuscript within 60 days; if you cannot complete the modification within this time period, please contact me. If you do not wish to modify the manuscript and prefer to submit it to another journal, please notify me of your decision immediately so that the manuscript may be formally withdrawn from consideration by Microbiology Spectrum.

Response to Reviewers

"Tezacaftor/Ivacaftor therapy has negligible effects on the cystic fibrosis gut microbiome" by Christopher van der Gast, Ryan Marsh, Claudio Dos Santos, Liam Hanson, Christabella Ng, Giles Major, Alan Smyth, and Damian Rivett [Paper #Spectrum01175-23]

Please note: All line numbers quoted in the responses now refer to the "Marked-up manuscript", unless specified as the original line number.

Reviewer #1 (Comments for the Author):

This is a nicely written and well-conceived report that should interest many in the cystic fibrosis community. It's a shame, of course, that Teza/Iva doesn't really do much to improve the intestinal microbiome and CF patient gut health. Maybe Elexa/Teza/Iva will.

Comment: By way of comments and suggestions, I would recommend following up on this statement on line 101: "Significant differences in diversity were observed in core taxa between baseline and placebo ($P = 0.007$) and placebo and Tezacaftor/Ivacaftor treatment groups ($P = 103\ 0.039$). I would mention that although these differences reach statistical significance, all three CF groups had values of Fisher's alpha of about .75 and observed differences would be of questionable biological significance.

Response: This is a valid suggestion, and as such the manuscript has been amended accordingly. Line 153 of the revised manuscript now reads:

"There were differences across the core taxa observed within the placebo treatment period, however the similar magnitude of Fisher's alpha likely indicates these changes render little biological significance and are resultant of alternate temporal variation."

Comment: Another suggestion I have is to add a sentence or two about why we might have predicted that Texa/Iva would reshape bacterial communities. For example, do you suspect that restoring chloride channel function in gut epithelial cells would improve gut hydration and thereby promote the growth of certain bacteria?

Response: This was somewhat addressed in the introduction. Line 75 originally read: "Given that the primary consequence of CFTR dysfunction alone is sufficient to induce dysbiosis in the CF population (19), it is also reasonable to suggest restoration of CFTR function could remodulate the bacterial composition back to a signature observed in healthy controls,".

However, the structure of the text has been altered emphasise this. We also have included in the discussion with mention of fluid transport but also potential other functions of the CFTR protein that could induce subsequent changes to the microbiota (10.3390/cells10112844) starting from line 72 in the revised manuscript:

"The CF gut microbiota is currently divergent from healthy controls throughout life and further compounded by CF-associated lifestyle factors such as antibiotic therapies and dietary changes (18–22). It is, however, reasonable to suggest restoration of CFTR function could remodulate the bacterial composition back to a signature observed in healthy controls, given that the primary consequence of CFTR dysfunction alone is sufficient to induce dysbiosis in the CF population (23). This may arise from the restoration of fluidity at the site of the intestinal epithelia, or through various other pathways and mechanisms in which the CFTR protein plays a key role (24)."

Reviewer #2 (Comments for the Author):

Thank you for the opportunity to review the submitted manuscript (Spectrum01175-23) by Ryan Marsh and colleagues, titled "Tezacaftor/Ivacaftor therapy has negligible effects on the cystic fibrosis gut microbiome". The study examined the impact of Tezacaftor/Ivacaftor treatment on the gut microbiota composition and metabolomic functionality in twelve people with cystic fibrosis (pwCF). The researchers partitioned bacterial taxa into common core and rare satellite taxa and compared diversity and composition for the whole microbiota, as well as the core and satellite taxa, between treatment periods and healthy controls. The study found that Tezacaftor/Ivacaftor treatment did not significantly affect the gut microbiota composition or diversity, nor did it impact the production of short-chain fatty acid (SCFA) metabolites. The study also found that the relative levels of acetic, propionic, and butyric acid were increased in pwCF compared to healthy controls, whereas longer SCFAs containing {greater than or equal to}5 carbons were more abundant in healthy controls. Finally, the study found no significant differences in intestinal inflammation or participant symptom scores between Tezacaftor/Ivacaftor and off-treatment samples. Overall, this study provides insight into the effects of Tezacaftor/Ivacaftor on the gut microbiota in pwCF. Overall, the study is scientifically and methodologically accurate and conclusion are well supported by the reported results. However, I find that the current study has some limitations, some that are already highlighted in the manuscript, that might require a bit more detail on prior the manuscript being considered suitable for publication in Microbiology Spectrum.

Points:

1. What about pwCF that did not receive Tezacaftor/Ivacaftor or any other modulator therapy? Would the trajectory in the gut microbiota composition be similar and relatively stable in pwCF that never have received modulator therapy? Is possible that maturation of the gut microbiome in pwCF is affected by early adaptation of the microbiota to the "abnormal" environment of the lower GI and high antibiotic pressure in general?

Response: It is an interesting discussion. The resilience of the CF gut microbiota has recently been demonstrated (<https://doi.org/10.1186/s12866-023-02788-y>), indicating that random perturbations to the community are likely drivers of change in adult life. The trajectory of the CF gut microbiota has been followed across children with CF and compared with controls, indeed indicating a role of antibiotics for this divergence from healthy controls (<https://doi.org/10.1016/j.jcf.2020.04.007>).

Whilst we think some of these points might lie outside the scope of what is necessary for the purpose of this manuscript, highlighting the recent reductions in administrable ages of CFTR modulator treatment may be relevant regarding microbiota maturation in pwCF, and has been added accordingly to the revised manuscript at line 227:

"It is also logical to postulate that the initiation of CFTR modulator treatment earlier in life will also increase microbiota similarity between pwCF and the wider population, particularly if other GI manifestations are minimised."

2. As this pilot study is relatively small, which limits the generalisability of the results. Is it possible that larger sample sizes could have revealed more significant differences in microbiota composition and SCFA metabolites?

Response: Current literature surrounding CFTR modulators and the gut microbiota encompasses mainly pilot studies, of which changes to species relative abundance and the wider microbiota structure have previously been demonstrated (<https://doi.org/10.1038/s41598-018-36364-6> & <https://doi.org/10.3390/jpm11050350>). It is therefore possible that with an increased sample size of pwCF the effects of Tez/Iva, regardless of their magnitude, will more confidently be realised. Both of these studies were following the use of Ivacaftor in patients harbouring class III mutations. It may be possible that such effects arise from increased efficacy of Ivacaftor for pwCF harbouring class III mutations, as compared to pwCF with p.Phe508del receiving Tez/Iva. The impact of Tez/Iva may therefore render subtle changes to the microbiota, of which a larger sample size will help elucidate. We have made the necessary changes to the revised manuscript to address this point on line 164:

“Given the increased efficacy of Ivacaftor for pwCF undergoing treatment for class III mutations relative to those administered dual-therapy to treat p.Phe508del in the respiratory domain (27), it may be possible that this reduced efficacy also extends to the site of the intestinal tract and changes to the microbiota are more subtle in such cohorts of pwCF. This will be clarified as further studies on CFTR modulators and the intestinal microbiome encompass greater numbers of participants.”

3. The study did not investigate the long-term effects of Tezacaftor/Ivacaftor on the gut microbiota, as the treatment period was only 4 weeks. Would the authors be able to further elaborate on if longer treatment durations could result in more significant changes in microbiota composition and SCFA or other metabolites in general?

Response: It is difficult to speculate whether longer-term treatment would result in significant changes to the microbiota as there is limited evidence of sustained usage of double-therapy CFTR modulators in pwCF harbouring the p.Phe508del mutation. We do in fact discuss changes to diversity and community composition from a study utilising Ivacaftor with extended sampling on line 159 in the original manuscript: “Whilst Kristensen et al did observe significant changes to the aforementioned (18), this was only evident following extended 12 months of treatment in pwCF harbouring the p.Ser1251Asn mutation. We do speculate at line 237 in the original manuscript: “The negligible effects observed in this study may be related to the short administration period of Tezacaftor/Ivacaftor alongside other characteristics of pwCF, including continuous antibiotic treatment and sustained pancreatic insufficiency.” Amendments have been to the revised manuscript for further clarity at line 239:

“Whilst the treatment period was also relatively short, longer-term administration has previously failed to elicit changes to the intestinal microbiota across pwCF with similar genotypes (16), however this does not include Tez/Iva administration.”

4. Studies have shown limited changes in airway microbiota following modulator therapy, despite clinical improvement (both for single [<https://doi.org/10.1513/AnnalsATS.201907-493OC>; <https://doi.org/10.1016/j.jcf.2020.12.023>], double therapy [<https://doi.org/10.1128/spectrum.02251-22>; <https://doi.org/10.1183/23120541.00731-2020>] or even triple therapy [<https://doi.org/10.1016/j.jcf.2021.11.003>]), could the author further elaborate on if we would have expected significant changes in the GI microbiota? Would this be more associated with putative changes in functional capacities members of the GI microbiota?

Response: It is reasonable to suggest that restoration of CFTR function alone could modulate significant changes to the microbiota composition, given primary consequence of CFTR dysfunction alone is sufficient to induce dysbiosis in the CF population (line 75, original manuscript). Whilst Tez/Iva is suitable for pwCF harbouring at least one copy of p.Phe508del, all participants in the current study were p.Phe508del homozygous and pancreatic insufficient. This has been previously associated with enhanced dysbiosis at the site of the GI tract as compared to pwCF harbouring less severe mutations (doi:10.1371/journal.pone.0061176). Cohorts in which microbiota changes have been observed are limited to Ivacaftor usage, which demonstrates an increased efficacy of action in those pwCF harbouring class III mutations such as p.Gly551Asp and p.Ser1251Asn (<https://doi.org/10.1038/s41598-018-36364-6> & <https://doi.org/10.3390/jpm11050350>).

Nonetheless, whilst the microbiota composition might not seemingly alter, as highlighted in the respiratory domain above, the integration of further -omic techniques will likely reveal if there are in fact changes to the functionality of keystone species that may positively impact clinical outcomes.

These responses have been collated with point 5 (below) and addressed in the revised manuscript at line numbers: 229-236: "Should this not arise, despite patient clinical improvement, the further integration of multi-omic approaches will likely clarify if the predisposed microbial community exhibit changes to functionality that promote a favourable intestinal environment for pwCF. Finally, ETI therapy demonstrates increased across intestinal epithelia as compared to Tez/Iva in biopsies from pwCF homozygous for the p.Phe508del mutation, which our current cohort all exhibited (49 As we obtain deeper knowledge surrounding triple-therapy CFTR modulator usage, the findings of this study should therefore contribute valuable insights to the complex challenge of comprehending the associations between CFTR functionality and alterations to the intestinal microbiota."

5. With many individuals now being moved onto triple therapy, would the authors be able to comment on this in context to the current study and the importance of their findings?

Response: As stated many pwCF are now moving onto triple-combination therapy, of which exhibits much improved efficacy in the respiratory domain of complications for pwCF harbouring the p.Phe508del mutation, as compared to the previous dual therapy options that were previously available (<https://doi.org/10.3390/ijms21165882>). Similarly, the effects upon the gastrointestinal tract are more promising, with increased CFTR function at the intestinal epithelia as compared to Tez/Iva administration alone in p.Phe508del homozygotes (<https://doi.org/10.1164/rccm.202110-2249OC>). Our work here will therefore help elucidate the impact of restoring CFTR-mediated intestinal current upon the microbiota composition in pwCF, should triple therapy have more pronounced effects upon the microbiota in future studies. If this is not to be the case, it will elaborate the need to further understand CF-

associated lifestyle effects and the impact of other CF manifestations upon the gut microbiota.

These responses have been collated with point 4 (above) and addressed in the revised manuscript at line number: 229-236: “Should this not arise, despite patient clinical improvement, the further integration of multi-omic approaches will likely clarify if the predisposed microbial community exhibit changes to functionality that promote a favourable intestinal environment for pwCF. Finally, ETI therapy demonstrates increased efficacy across intestinal epithelia as compared to Tez/Iva in biopsies from pwCF homozygous for the p.Phe508del mutation, which our current cohort all exhibited (49). As we obtain deeper knowledge surrounding triple-therapy CFTR modulator usage, the findings of this study should therefore contribute valuable insights to the complex challenge of comprehending the associations between CFTR functionality and alterations to the intestinal microbiota.”

June 28, 2023

Prof. Christopher van der Gast
Northumbria University
Department of Applied Sciences
Ellison Building
Newcastle upon Tyne NE1 8ST
United Kingdom

Re: Spectrum01175-23R1 (Tezacaftor/Ivacaftor therapy has negligible effects on the cystic fibrosis gut microbiome)

Dear Dr. van der Gast:

I'm pleased to tell you that your manuscript has been accepted to Microbiology Spectrum.

Sincerely,
Joanna

I am forwarding it to the ASM Journals Department for publication. You will be notified when your proofs are ready to be viewed.

Sincerely,

Joanna Goldberg
Editor, Microbiology Spectrum
